# Comparative Mitogenomics Analysis Revealed Evolutionary Divergence among *Neopestalotiopsis* Species Complex (Fungi: *Xylariales*)

**DOI:** 10.3390/ijms25063093

**Published:** 2024-03-07

**Authors:** Yutao Huang, Huanwei Wang, Siyan Huo, Jinpeng Lu, Justice Norvienyeku, Weiguo Miao, Chunxiu Qin, Wenbo Liu

**Affiliations:** Key Laboratory of Green Prevention and Control of Tropical Plant Diseases and Pests, Ministry of Education, School of Tropical Agriculture and Forestry, Hainan University, Haikou 570228, China; 21220951320064@hainanu.edu.cn (Y.H.); 20090401210010@hainanu.edu.cn (H.W.); 22220951320059@hainanu.edu.cn (S.H.); 23220951320084@hainanu.edu.cn (J.L.); jk_norvienyeku@hainanu.edu.cn (J.N.); miao@haianu.edu.cn (W.M.)

**Keywords:** *Neopestalotiopsis*, mitogenome, tRNA, speciation, phylogenomics

## Abstract

The genus *Neopestalotiopsis* consists of obligate parasites that cause ring spot, scab, and leaf blight diseases in higher plant species. We assembled the three complete mitogenomes for the guava fruit ring spot pathogen, *Neopestalotiopsis cubana*. The mitogenomes are circular, with sizes of 38,666 bp, 33,846 bp, and 32,593 bp. The comparative analyses with *Pestalotiopsis fici* showed that *N. cubana* differs greatly from it in the length of the mitogenomes and the number of introns. Moreover, they showed significant differences in the gene content and tRNAs. The two genera showed little difference in gene skewness and codon preference for core protein-coding genes (PCGs). We compared gene sequencing in the mitogenomes of the order *Xylariales* and found large-scale gene rearrangement events, such as gene translocations and the duplication of tRNAs. *N. cubana* shows a unique evolutionary position in the phylum *Ascomycota* constructed in phylogenetic analyses. We also found a more concentrated distribution of evolutionary pressures on the PCGs of *Neopestalotiopsis* in the phylum *Ascomycota* and that they are under little selective pressure compared to other species and are subjected to purifying selection. This study explores the evolutionary dynamics of the mitogenomes of *Neopestalotiopsis* and provides important support for genetic and taxonomic studies.

## 1. Introduction

Mitochondria are critical organelles responsible for crucial processes in eukaryotic cellular function. They oversee asymmetric transport protocol (ATP) generation through oxidative phosphorylation, the synthesis of vital metabolites, and iron storage [1]. The evolutionary origins of eukaryotes from an endosymbiotic α-proteobacterium profoundly influence their evolutionary trend. These ancient associations are evident in contemporary eukaryotes, demonstrated by autonomous mitochondrial DNA (mtDNA) or nuclear genes of *α-proteobacterial* origin [2]. 

Mitogenomes acts as a crucial genetic entity that resides in the mitochondria of eukaryotic cells, offering significant insights into the processes of evolution. Mitogenome-related investigations across different species revealed a considerable diversity, allowing for a sturdy foundation for taxonomic clarity [3]. The high copy numbers and evolutionary rates of fungal mtDNFA make it an exceptional tool for evolutionary biology, taxonomy, and molecular phylogenetics [4,5,6,7,8,9]. Mitochondrial genomes possess distinct characteristics, such as a condensed size, constrained recombination rates, autonomous evolutionary path, and the abundance of analogous genes, which equip them with remarkable analytical capabilities when examining the subjects of population genetics and evolutionary biology [10,11,12,13].

With the emergence of next-generation sequencing technology, it has become easier to sequence mitochondrial genomes, resulting in a vast collection of complete mtDNA sequences across a variety of taxa, including animals, plants, and fungi [14]. In recent years, there has been a notable increase in research focused on fungal mitochondrial genomes, revealing various intricate findings. Notably, these investigations have uncovered significant variations in genome sizes, repetitive patterns, sequences between genes, intron make-up, code sequences, and genomic organization in different fungal taxa [13,15,16]. Recent studies have shed light on the complex characteristics of fungal mtDNA and have delved into uncovering significant evolutionary connections, enhancing our understanding of eukaryotic phylogenetic relationships more broadly. Research on fungal mitogenomics has also covered functional genomic discussions, revealing the crucial functions of mtDNA-encoded genes in vital cellular processes.

Moreover, recent academic studies have investigated the evolving factors of mitochondrial evolution, encompassing the causations of genome-size discrepancies and the impact of horizontal gene transfer occurrences [17,18,19]. The progressive generation fungal mitochondrial genomic resources will provide resource support, enhance insights into fungal biology, and significantly accelerate research efforts to provide a comprehensive understanding of eukaryotic evolution. However, investigations on fungal mitochondrial genomes still need to be more extensive in contrast to those on animals and plants.

Members in the Neopestalotiopsis genus fall under different families, including *Sporocadaceae*, the *Amphisphaeriales* order, and the *Sordariomycetes* class [20]. The *Neopestalotiopsis* species have a broad ecological range, as they function as saprobes, tree associates, and post-harvest disease agents, typically causing fruit ring spot and leaf blight diseases. The genus has a worldwide distribution that transcends geographic limitations, with documented occurrences in astonishing habitats, including subterranean environments like caves in China [21,22,23]. Public repository databases have no comprehensive mitochondrial genome for members within the *Neopestalotiopsis* species complex. Relatively, the closest phylogenetic relative of the *Neopestalotiopsis* species with a complete mitochondrial genome resource deposited in the National Center for Biotechnology Information (NCBI) database (https://www.ncbi.nlm.nih.gov/, accessed on 2 March 2021) belongs to the *Pestalotiopsis* genus [19].

In this paper, we set out to generate high-quality mitochondrial genome assemblies for three *N. cubana*, perform comparative inter- and intra-species mitogenomic analyses, and define the systematic position of *N. cubana* within the order *Ascomycota*. Owing to the absence of a reference mitogenome assembly for members in the *Neopestalotiopsis* genus, the publicly available mitogenome resource of members in the *Pestalotiopsis* genus [19] was adopted and deployed for comparative inter-species mitogenomic analysis. This study will significantly enhance our understanding of the evolution of the *Neopestalotiopsis* genus and enrich the existing mitochondrial genome information for this species. It will provide valuable references for future studies of the mitochondrial genome’s organizational structure and the gene sequence of this species.

## 2. Results

### 2.1. Structure and Organisation of the Mitochondrial Genome of N. cubana

We generated annotated mitochondria genome assemblies for three *N. cubana* field isolates obtained from three major guava cultivation zones within the Hainan Province, Wuzhishan, Ledong, and Baoting. The mitogenomes of the three isolates displayed high similarities, with recorded lengths of WZS16 (38,666 bp), LD08 (33,846 bp), and BT03 (32,593 bp) (Figure 1). The mitogenomes obtained for the three isolates showed a circular structure, with each containing 14 core PCGs and 1 ribosomal protein S3 (rps3) gene. Characteristically, all the genes are on the J strand. However, discrepancies exist in the number of open reading frames (ORFs) listed in Appendix A. Upon comparing the mitochondrial genomes of *Pestalotiopsis fici* W106-1 to those of the four organisms identified within the *Xylariales* order, it is apparent that these genomes exhibit a comparable GC content (from 27.48 to 28.45) and tRNA counts (from 31 to 34). Furthermore, we showed that the GC-skew values of *N. cubana* and *P. fici* W106-1 ranged from 0.107 to 0.118, while the AT-skew values ranged from −0.040 to 0.007. These values exhibited a remarkable similarity. Also, we recorded an overlap in the nucleotide positions of the neighboring genes nad4L and nad5 (−1 bp) in the mitogenomes of *N. cubana* and *P. fici* W106-1.

Additionally, we demonstrated that protein-coding regions occupied the highest proportion (40.80%, 40.80%, and 56.92%) of the mitogenomic region of the individual strains (Figure 2). Ribosomal RNA (rRNA) sequences (tRNA and rRNA) occupied the lowest (from 6.94% to 19.30%) proportion of the mitogenomic regions. The proportion of intron sequences identified in the mitogenome varied significantly between the individual strains. In the mitochondrial genomes of the *N. cubana* strains and *P. fici* W106-1 strains, the percentage of intronic sequences recorded was 0% (*N. cubana* WZS16 and *N. cubana* BT03), whereas the percentage of introns in *P. fici* W106-1 reached 28.77% (Figure 2). The single intron region identified exclusively in the Ledong isolate accounted for 3.60% of its mitogenome. By comparing the lengths of the three mitochondrial genomes from Hainan Province to that of *P. fici* W106-1, it is apparent that the number of introns is a significant factor in determining the size of mitochondrial genomes. Moreover, the proportion of intergenic sequences in the four mitochondrial genomes is strikingly similar, ranging from 23.49% to 25.92%, indicating the loose structure of their mitochondrial genomes.

### 2.2. Codon Usage Analysis

In the protein-coding genes (PCGs) of these four mitochondrial genomes, ATG functions as the start codon, and most of the stop codons are TAA, except for the cob, nad4, and nad5 genes, which use TAG as the stop codon (Appendix A). Also, the cob genes recorded in the mitogenomes of the three *N. cubana* field isolates possess a TAA-type stop codon, while the nad5 gene in the mitochondrial genome of *P. fici* W106-1 also uses TAA as its stop codon.

The codon usage and relative synonymous codon usage (RSCU) in the four mitogenomes were analyzed. The results reveal a high similarity in the codon usage pattern between *N. cubana* and *P. fici* W106-1. AGA (which encodes Arg2) and UUA (which encodes Leu2) represented the most frequent codons used (Figure 3a–d), with both amino acids being the most abundant. Moreover, out of the 62 codons analyzed, 26 had RSCU values above 1.0, with A and U occurring particularly often.

### 2.3. RNA Genes in the N. cubana Mitogenomes

We identified the sequences of two genes coding for the large subunit ribosomal RNA (rnl) and small subunit ribosomal RNA (rns) in the mitogenomes of the three *N. cubana* isolates and the *P. fici* W106-1 strains. All the mitogenomes analyzed in this study contained 31 to 34 tRNA-coding genes. All the recorded mitogenomic tRNA genes were condensed into the conventional cloverleaf structure (Figure 4). These tRNAs encoded the 20 standard amino acids. Each of these mitogenomes contained specific tRNAs that utilized distinct anticodons to code for amino acids encoding arginine, leucine, serine, and glycine (Appendix A). Furthermore, three tRNAs encoded methionine in the individual mitogenomes and had the same anticodon. Of the thirty tRNAs shared by the mitogenomes *N. cubana* isolates and the *P. fici* W106-1 strains, four tRNAs possessed extra arms that were much longer than the others. The results suggest a variation in the size of tRNA due to the variations in the size and length of the extra arms of the four mitogenomic tRNAs. Within these thirty shared tRNAs, a total of fifty-six variable sites were designated, which they appear in fifteen tRNAs. The most frequent appearance is trnG (tcc), with a total of thirty variable sites identified.

### 2.4. Repetitive Sequences in the Mitogenome

We analyzed scattered repeat sequences in the three *N. cubana* mitogenomes. The results show that the mitochondrial genomes of *N. cubana* from the Ledong region of Hainan Province, China, contained 131 forward types (3836 bp in total), 54 reverse repeats (1219 bp in total), 31 complement types (689 bp in total), and 78 palindromic repeats with lengths sizes of 22 bp or more (a total of 1778 bp). The remaining two scattered repeats in the mitogenome were comparable (Appendix A), indicating that most of the scattered repeats are within the intergenic region of the mitogenomes (Figure 5). A further examination of tandem repeat sequences in the mitogenomes of the three *N. cubana* isolates and the *P. fici* W106-1 strains with the tandem repeats finder identified four tandem repeat sequences across the mitogenomes of the three *N. cubana* isolates. The sequence fragments of these tandem repeats are similar in the mitogenome of the individual *N. cubana* isolates, with the length of the longest tandem repeat sequence being 54 bp, while the length of the shortest tandem repeat sequence was 3 bp (Appendix A). In contrast, the mitochondrial genome of *P. fici* W106-1 contained 19 tandem repeats ranging from 3 to 45 bp in length. The MISA online tool-assisted analysis of the mitogenomes of *N. cubana* isolates and the *P. fici* W106-1 strains revealed the presence of simple sequence repeats (SSRs) in the individual mitogenomes (Appendix A). We observed that the predominant proportions of the detected SSRs consisted of single-nucleotide repeats with lengths in the range of 11–28 bp. The remaining proportion of the less frequent types of SSRs recorded included dinucleotide repeats (from 0 to 2), trinucleotide repeats (from 4 to 7), tetranucleotide repeats (from 6 to 9), pentanucleotide repeats (from 0 to 2), and hexanucleotide repeats (from 2 to 3).

### 2.5. Gene Rearrangement in the Mitochondrial Genome

The mitogenomes of the three *N. cubana* isolates was used for further comparative mitogenomic analysis against additional members from the order *Xylariales*, including Annulohypoxylon stygium, *Nemania diffusa* (GenBank accession number MH620794.1, NC_049077.1), and *Arthrinium arundinis* (GenBank accession number NC_035508.1). The examinations revealed minimal mitogenomic differences in the orientation of protein-coding genes, rRNA-coding genes, and tRNA-coding genes between the *N. cubana* isolates and *P. fici* W106-1. However, there were substantial differences in the orientation of protein, rRNA, and tRNA-coding genes between the mitogenomes of *N. cubana* and species from distantly related orders (Figure 6). While no differences were recorded in the orientation of genes between the three *N. cubana* isolates, we observed that the trnG gene in the mitogenome of *N. cubana* isolates from Wuzhishan experienced a doubling event. 

At the same time, we found differential doubling-associated differences in the trnC gene and the number of trnG genes between the mitogenomes of *N. cubana* and *P. fici* W106-1. For instance, besides the difference in the doubling pattern of the trnC gene, one of the trnG genes was absent in the mitogenome of *N. cubana* compared to that of *P. fici* W106-1. Incidences of inversions (trnS and trnR genes), deletions, and doubling (trnK and trnC genes) were recorded in the mitogenome of *Annulohypoxylon stygium.* However, the pattern of these inversions, deletions, and doublings differs from the pattern observed in the mitogenome of *N. cubana*. On the other hand, the mitochondrial genomes of *Nemania diffusa* and *Arthrinium arundinis* underwent significant gene rearrangements, indicating frequent gene rearrangement events during the evolution of species belonging to the *Xylariales* order.

Using the Mauve 2.4.0 software, we analyzed seven mitogenomes for the covariance of comparable gene regions and found that *N. cabana’s* mitochondrial genome had five homologous regions in common with other fungi in the *Xylariales* order (Figure 7). The positions of these homologous regions showed a significant variation amongst the different mitochondrial genomes. This is consistent with the previously mentioned gene order comparison, further supporting the occurrence of gene rearrangement in the mitochondrial genomes of the *Xylariales* order during evolution. While the size of the homologous regions varies substantially across different mitochondrial genomes, the mitochondrial genomes of *N. cubana* exhibit minor differences in these regions that are consistent with the lengths of their mitochondrial genomes. This could be ascribed to intergenic regions at the corresponding positions.

### 2.6. Variations in Core PCGs

In the examined mitogenomes of *N. cubana* and *P. fici W106-1*, the 14 core PCGs and rps3 were almost identical in length or GC content (Figure 8A), with only minor differences in gene length for cox1 and rps3, and no change in the GC content for the three *N. cubana* in the fold plot. This observation implies that core PCGs are conserved during the evolution of the *Sporocadaceae* family. Of the 15 core protein-coding genes analyzed, the atp9 gene exhibited the highest mean GC content of 35.11%, while the cox3 gene was a close second at 32.35% (Figure 8B). Conversely, the nad6 gene showed the lowest GC content, with a mean of 20.45%. The AT skew or GC skew values obtained for these 15 genes varied to some extent across the mitochondrial genomes of the four *Sporocadaceae* families. The only gene with a negative skew in the AT skew was rps3 (Figure 8C). All genes except for atp6 and atp8 skewed positively in the GC skew (Figure 8D). This result implies that most of the leading strands of the core PCGs have a bias towards T-rich and G-rich evolution during evolution.

### 2.7. Phylogenetic Analysis

To clarify the evolutionary connections of the *N. cubana* genus within the *Ascomycota* phylum, we selected 169 mitochondrial genomes of *Ascomycota* fungi from the NCBI database. All of these fungi have been identified as saprophytic and plant-pathogenic. PCGs from the 14 cores of these mitochondrial genomes were employed in the joint construction of a phylogenetic tree using maximum likelihood (ML) methods. The replacement model, with types compatible with multi-gene aligned partitions, was adopted and plotted as a map (Figure 9), with *Phytophthora infestans* used as an outgroup, for the evolutionary and phylogenetic evaluation of the relationship among these 169 genera using the complete length of all mitochondrial genomes. The results reveal 11 significantly diverse evolutionary clades with strongly support (BS > 83) (Figure 9).

Additionally, we observed that taxa within the same phylogenetic clades exhibited parallel mitochondrial genome lengths. The mitochondrial genomes of three *N. cubana* were clustered and closely related to *P. fici* W106-1, the most prevalent mitochondrial genome of *N. cubana*. This finding supports the conclusions of previous studies on the phylogenetic relationships of *P. fici*. This study provides further evidence for the effectiveness of the mitochondrial genome as a molecular marker in speciation. It can be utilized for analyzing the phylogenetic relationships among species within the phylum *Ascomycota*.

### 2.8. Evolutionary Adaptation of Core PCGs

The intensity of selection pressure on the mitochondrial genomes of *N. cubana* and other genera was assessed against more than three species from the *Ascomycete* phylum by examining multiple selection-prone parameters, including the non-synonymous substitution rate (dN) and synonymous substitution rates (dS). The ratio of dN/dS was obtained by extrapolation against the 14 fungal mitochondrial genomes of core PCGs of fungi species from 15 genera. The peak distribution values recorded from the 15 genera were mapped using computed average dN/dS. Interestingly, the observed dN/dS distributions fell into two main categories. The first group comprised species from five genera and included *Cladosporium*, Clonostachys, Colletotrichum, Epichloe, Fusarium, and Lecanora, and *Neopestalotiopsis*, Penicillium, Rhynchosporium, and *Trichoderma* showed positive dN/dS values with narrow peak distributions. The second group comprised species of the remaining 10 genera, with a relatively flat dN/dS distribution (Figure 10A). The further intra-species evaluation of the two *Neopestalotiopsis* species, *N. cubana* and *Pestalotiopsis fici* from the family *Sporocadaceae*, revealed dN and dS values in the ranges of (0–0.03) for *N. cubana* and (0.12–0.43) for *P. fici*. The highest dN/dS value of (0.03) was recorded for nad5, while atp8, atp9, nad1, nad3, and nad4L recorded the lowest, with a dN/dS value of 0. These results suggest the core PCGs in *N. cubana* and *Pestalotiopsis fici* W106-1 experience an intense purifying selection during the evolutionary process (Figure 10B). As for the evaluation of dN and dS values, most of *Lecanora’s* PCGs had the highest dN or dS values among all genera. Furthermore, *Bipolaris*, *Ceratocystis*, *Chrysoporthe*, *Cladonia*, *Cladosporium*, *Clonostachys*, *Epichloe*, *Erysiphe*, *Fusarium*, *Neopestalotiopsis*, and *Rhynchosporium* contained a total of 36 core PCGs, all of which had a dN value of 0. These results indicate that the environmental evolutionary pressures to which these species are subjected are largely biased in favor of purifying selective action. However, there were some taxa of species with slightly higher dN values, such as atp8 and nad3 in *Chrysoporthe* and nad6 in *Penicillium*, and their values ranged from 0.07 to 0.16, which indicates that they are under slightly higher environmental pressures on specific core protein-coding genes than other genera. Meanwhile, the recorded dS values for individual species in the 15 genera were higher than 0, with nad5 in *Epichloe* recording the lowest dS value of (0.002). Most of the genes in *Colletotrichum*, *Penicillium*, and *Lecanora* had dS values above 0.5, and *Lecanora’s* atp6 and nad4 tended to have dS values of 2.0, which means that their genes have a high rate of mutation but do not affect the amino acid sequences to be significantly changed. It is worth mentioning that the nad3 gene of *Chrysoporthe* has high dN and dS values, which may indicate that the nad3 genes of this taxon play different roles in different environments and are subject to complex environmental selection pressures. At the same time, the results obtained from the evaluation of the dN, dS, and dN/dS values for PCGs for species sampled from the 15 genera show that the dN/dS value of nad4L in *Bipolaris* is the highest among all the PCGs, and the dN/dS values of its genes are also higher than those of the other genera, as well as cox3 and cob, showing that they evolved faster in the fungal genus. None of the taxa had dN/dS values exceeding 1, with the largest being nad4L in *Bipolaris*, indicating that they experienced purifying selective effects. Overall, these genes are evolutionarily stable, indicating that they play important and conserved functional roles in the mitochondrial genome.

## 3. Discussion

Generally, the organization of the mitochondrial genomes of closely related species, the secondary structure of tRNAs, codon usage preferences, the location and number of repetitive sequences, the phenomenon of gene rearrangements, the core protein-coding genes, and the selective pressures exerted during the evolutionary process are important to understand the origins of evolution and the phylogenetic relationships of related species. Compared to plants and animals, there are limited mitogenomic studies in fungi, especially species within the phylum *Ascomycota*. However, although data on fungal mitochondrial genomes has increased considerably in recent years, studies have yet to be carried out on the mitochondrial genome of *N. cubana*. Before this study, the mitochondrial genomes of the *Sporocadaceae* family were available for only one species, *P. fici* W106-1. We generated the first mitogenome assembly resources for three *N. cubana* field isolates, the first mitochondrial genome of *N. cubana*, and the second mitochondrial genome of the family *Sporocadaceae* to be published [24].

We assembled three mitochondrial genomes of *N. cubana* with lengths of 38,666 bp, 33,846 bp, and 32,593 bp from Wuzhishan, Ledong, and Baoting in Hainan Province, China, respectively. We found that the mitochondrial genome of *N. cubana* was smaller than that of most other fungi, and we speculate that the main reason for this is that the mitochondrial genome has too few introns, with only one intron in each of the three mitochondrial genomes. The research findings show that *Pestalotiopsis* and *Neopestalotiopsis* are very closely related because *Neopestalotiopsis* was previously considered a separate genus from *Pestalotiopsis*, and their evolutionary direction and the changes that occurred must be very similar, and this study further confirmed this. In the comparison between *N. cubana* and *P. fici* W106-1, the major differences were in the number of introns and the length of the mitochondrial genome itself, which were much greater in *P. fici* W106-1 than in *N. cubana*. *These* results are consistent with previous suggestions that introns constitute critical factors that determine mitogenome size. These are some of the most important factors affecting the size of the mitochondrial genome, as found in previous studies. The number of ORFs in these four mitochondrial genomes differed, especially in the Wuzhishan region, where the mitochondrial genome of *N. cubana* was much larger than that of the other two regions, whereas the number of ORFs in *P. fici* W106-1 was similar to that of *P. fici* W106-1; so, we speculate that the number of ORFs is not a key factor in determining the differences in mitochondrial genome size. In addition, the intergenic sequences also varied greatly among these four mitochondrial genomes, and the percentage of intergenic sequences was to some extent directly positively correlated with the length of the mitochondrial genomes, which is consistent with the conclusion that intergenic sequences are also a factor affecting the size of mitochondrial genomes in previous studies.

Codon usage biases play a crucial role in protein function and translation accuracy, and codon usage analyses can help to study evolutionary and environmental adaptations in different species. We assessed codon usage preference in the mitochondrial genomes of *N. cubana* and *P. fici* W106-1 and found that the codon biases in the mitochondrial genomes of the two species were almost identical. The start codons of the core protein-coding genes of the four mitochondrial genomes were also identical, whereas the termination codons were used slightly differently. A total of 26 out of 62 codons had RSCU values greater than 1.0, particularly AGA (coding for Arg2) and UUA (coding for Leu2), likely due to the impacts of environmental stresses.

TRNAs act as transporters and play a crucial role in protein translation. Research demonstrations showed that changes in the number of extra arms cause changes in tRNA length between the mitochondrial genomes of closely related species. In tandem with previous observations, the current investigation showed that, of the 30 tRNAs common to the four mitogenomes, 4 tRNAs possessed extra arms that were much longer than the others. Variable sites in the secondary structure of tRNAs partly reflect the prevailing evolutionary differences between the species but have no impact on the transporter function of the tRNAs themselves. Interestingly, we identified 56 variable sites in these 30 common tRNAs, which were present in 15 tRNAs, of which trnG (tcc) was the most abundant, with 30 variable sites. From these observations, we speculated that the use of glycine in *N. cubana* and *P. fici* W106-1 underwent a relatively different evolutionary process due to environmental stresses, leading to a frequent horizontal variation in their corresponding tRNAs.

In early studies, the order of genes in fungal mitochondrial genomes was relatively conservative because they all came from the same ancestor, *α-proteobacterium*. However, with the increasing spate of fungal mitogenome resources, detailed insights on crucial mitogenomic features, including gene rearrangement, are beginning to emerge. Studies have demonstrated that the accumulation of repetitive sequences in fungal mitochondrial genomes causes dynamic changes in the genome structure during evolutionary processes, leading to gene order rearrangement. We detected different repetitive sequences in the mitochondrial genomes of *N. cubana*. We analyzed lengths, sizes, and positions in the genomes. The distribution of repetitive sequences in the three mitochondrial genomes was the same in position and length. Taking the mitochondrial genome circular map of *N. cubana* in Ledong as an example, it can be seen that a considerable proportion of the scattered repetitive sequences (≥22 bp) are on intergenic sequences, which can be corroborated with the gene rearrangement phenomenon of *N. cubana* and *P. fici* W106-1. Similarly, tandem and SSRs were mostly on intergenic sequences. The comparison of the gene sequences with the mitochondrial genomes of other *Xylariales* species and the covariance analysis showed that their gene rearrangements became more diverse, suggesting that gene rearrangements in mitochondrial genomes occur very frequently during the evolutionary process and, to some extent, reflect the phylogenetic relationships of the species.

Core protein-coding genes are well conserved in mitochondrial genomes and show slight variations early in the evolutionary process. With time and environmental changes, they tend to undergo either positive or purifying selection, and differences in core PCGs in different mitochondrial genomes may also reflect the rate of evolution. By comparing the core PCGs of *N. cubana* and *P. fici* W106-1, we found that the differences in length, GC content, AT skew, and GC skew between the two species were minimal, and we speculate that the rate of core PCG evolution is the same in species below the family level. The analyses of dN value, dS value, and dN/dS of the core PCGs of *N. cubana* and *P. fici* W106-1 also reflected that their PCGs underwent a strong purifying selective effect during the evolutionary process. The core PCGs of 169 species of *Ascomycetes* showed different evolutionary directions over time, with the dN/dS distributions mainly divided into categorical and narrow peaks and flat distributions. The dN/dS of *N. cubana* was almost the smallest in *Ascomycota*, indicating that its evolutionary rate is relatively slow in the whole *Ascomycota* system, which may be related to the environmental status of the genus *Neopestalotiopsis* in Hainan Province, China.

*Neopestalotiopsis* was originally placed in the genus *Pestalotiopsis*, but currently, *Neopestalotiopsis*, Pseudopestalotiopsis, and Pestalotiopsis are all placed in the family *Sporocadaceae*, which has undergone numerous taxonomic changes since its inception. This phenomenon likely accounted for the difficulties encountered in accurately classifying the fungi of *Neopestalotiopsis* based on morphological characteristics alone. The mitochondrial genome is becoming an important factor in the analysis of the phylogenetic relationships of fungi. Although phylogenetic analyses based on the nuclear genome provide richer genetic information, the nuclear genome is too expensive to obtain, and the manpower and resources required are much higher than those for the mitochondrial genome, making it difficult to obtain the nuclear genome of fungi in bulk and in large quantities within a short period. To further validate the position of *Neopestalotiopsis* in the phylogenetic relationships of fungi, we assembled 14 nuclear protein-coding genes in the mitochondrial genomes of 169 species in the phylum *Ascomycetes* for phylogenetic analyses and constructed phylogenetic trees using the maximum likelihood (ML) method. It can be seen that fungi from all three regions of the *N. cubana* cluster on the same branch with a high degree of support. *P. fici*, on the other hand, is closely related to *N. cubana* and is the closest to other fungi in the order *Xylariales*. These results agree with those of previous studies of phylogenetic relationships within the *Ascomycetes* phylum and provide further evidence that the mitogenome represents a powerful tool for analyzing phylogenetic relationships in the fungal community.

In this study, we assembled the mitochondrial genome of *N. cubana* and furthered our understanding of fungal mitochondrial genomes by comparing them to other closely related species. However, access to the mitogenomes of fungi in the order *Xylariales* and family *Sporocadaceae* is limited and inadequate to support large-scale mitogenome phylogenomic profiling. In the future, as more mitochondrial genomes are sampled, these analyses will become more valuable in population genetics, taxonomy, and evolutionary biology studies of fungi.

## 4. Materials and Methods

### 4.1. Sample Collection and DNA Extraction

Field isolates of the causal pathogen of fruit ring spot disease in guava were collected from orchards in Wuzhishan (109.29° E, 18.56° N), Ledong (108.51° E, 18.32° N), and Baoting (109.32° E, 18.35° N) in Hainan Province of China. The strains were collected from the field and isolated and cultured, and then their pathogenicity was determined by single-spore isolation. Then, we observed the conidial morphology of the fungi, and finally, through polymerase chain reaction of three nuclear genes (internal transcribed space, tubulin-β, and transcription elongation factor) and a BLAST comparison in the NCBI database, we confirmed that the three genes of these fungi all belonged to the same species, and thus the results of the co-identification were obtained. The fungus was isolated and identified as *N. cubana*, with the numbers of the fungus in the order of WZS016, LD08, and BT03. Preservation facilities at the School of Tropical Agriculture and Forestry, Hainan University, were used to preserve the samples. After 3 days of cultivation, we collected the mycelium from the liquid potato glucose culture medium. Total genomic DNAs were extracted from the isolated fungal samples using the Pathogenic Microbiome DNA Kit (CW3030 # CWBIO Jiangsu, China), according to the manufacturer’s instructions.

### 4.2. Sequencing, Assembly, and Annotation

The NexteraXT DNA Library Preparation Kit was used to construct libraries with an average length of 350 bp. The reads were generated using the Illumina Novaseq 6000 sequencing platform, and the average length of the generated reads was 150 bp. The Illumina-based raw reads were edited using the NGS QC Tool Kit v2.3.3 [25]. When the quality of the sequencing data was qualified, a complete circular assembly graph was constructed and further extracted by the visualization (e.g., Bandage) of the GFA graph files that were assembled using the SPAdes 3.14.1 software [26,27]. We used MITOS (http://mitos.bioinf.uni-leipzig.de/index.py, accessed on 4 March 2022) and MFannot (https://megasun.bch.umontreal.ca/cgi-bin/mfannot/mfannotInterface.pl, accessed on 6 June 2022) to annotate genes based on genetic code 4 [28]. Combining the results of the two methods of annotation, the annotated genes were then manually corrected for start and stop codons, and the corrected, annotated genes were compared using BLASTn for secondary correction. A subsequent comparative mitogenomic analysis between *N. cubana* and species from closely related genera was conducted. We modified the initial annotation to include the protein-coding (PCGs), rRNA, and tRNA genes. Additionally, the predicted protein-coding genes (PCGs) were further validated with the NCBI open box search tool (https://www.ncbi.nlm.nih.gov/orffinder, accessed on 10 June 2022) in conjunction with blastp and the NCBI non-redundant protein sequence database [29]. Exonatic V2.2 was employed to verify the inner content–outer sub-border of PCGs [30]. We also used tRNAscan-SE v1.3.1 to predict the tRNA gene [31]. The circular mitochondria genome map was constructed using Organellar Genome DRAW v1.2 [32].

### 4.3. Comparative Analysis of the Mitogenomes of a Variety of Fungi

Three mitochondrial genomes of the fungus *N. cubana* and the fungus *Pestalotiopsis fici* W106-1 were chosen for analysis to compare their size and structure. The AT-skew (=A−T/A + T) and GC-skew (=G−C/G + C) were computed to determine the skewness of the base content. The secondary structures of the tRNA-coding genes were predicted with the MITOS Web Server (http://mitos.bioinf.uni-leipzig.de/index.py, accessed on 11 June 2022). A codon usage analysis was performed using the codon W1.4.2 software for prediction. The simple sequence repeats (SSRS) were determined using the MISA online tool (https://webblast.ipk-gatersleben.de/misa/index.php?action=1, accessed on 11 June 2022), and tandem Repeats Finder (http://www.repeatmasker.org/, accessed on 11 June 2022) was used to recognize tandem repeats [33]. REPuter (https://bibiserv.cebitec.uni-bielefeld.de/reputer, accessed on 11 June 2022) was used to identify forward, reverse, complementary, and palindromic interspersed repeat sequences, and Tbtools was used to generate a circos map [34,35]. We chose the Mauve 2.4.0 software to analyze the gene rearrangements for collinearity analysis [36].

### 4.4. Phylogenetic Analysis

In order to ascertain the evolutionary relationships between *N. cubana* and other filamentous *Ascomycetes*, the mitochondrial genome sequences and annotation data for selected filamentous *Ascomycetes* were retrieved from the NCBI database, with *Phytophthora infestans* (NC_009905.1) serving as the outgroup. The mitochondrial genomes of these fungi’s 14 core PCGs (apt6, atp8, atp9, nad1, nad2, nad3, nad4, nad4L, nad5, nad6, cox1, cox2, cox3, and cob) were individually aligned by MAFFT v7.505 [37]. The results from the alignment of the protein-coding genes (PCGs) were merged using SequenceMatrix v1.7.8, followed by a determination of the optimal model fit for each PCG using ModelFinder. The maximum likelihood (ML) method was used to construct phylogenetic trees. The ML analyses were evaluated by bootstrapping with 1000 replications using IQ-Tree v2.2.0.3. Four chains (three hot and one cold) were run for a total of 10 million generations, with a sampling rate of 100 generations. The phylogenetic trees were pruned and stylized using iTOL v6 [38,39].

### 4.5. Availability of Data

The *N. cubana* mitogenome sequences were deposited in the GenBank under the accession numbers NC071220.1, OQ707026.1, and OQ707025.1, and the associated BioProject, SRA, and Bio-Sample numbers are PRJNA857506; SRR20075021, SRR23999140, and SRR23998960; and SAMN29628486, SAMN33958487, and SAMN33958289, respectively.

## Figures and Tables

**Figure 1 ijms-25-03093-f001:**
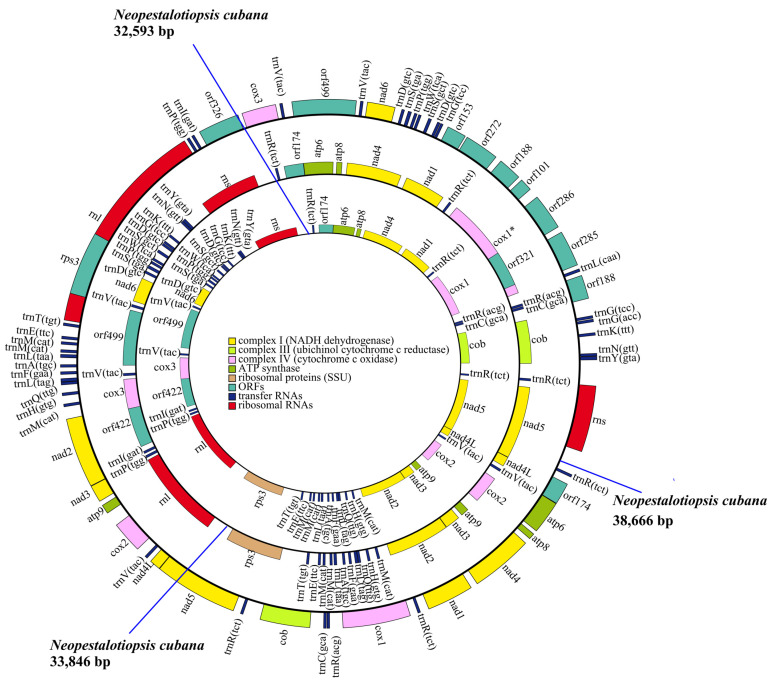
The three circular graphs represent the complete mitochondrial genomes of *N. cubana* from different areas. Different genes are represented with different color blocks. The color blocks outside the ring indicate that all genes are oriented in the same direction. The rings from the inside to the outside represent the mitochondrial genomes of different regions: Baoting area, Hainan Province, China (32,593 bp); Ledong area, Hainan Province, China (33,846 bp); and Wuzhishan area, Hainan Province, China (38,666 bp).

**Figure 2 ijms-25-03093-f002:**
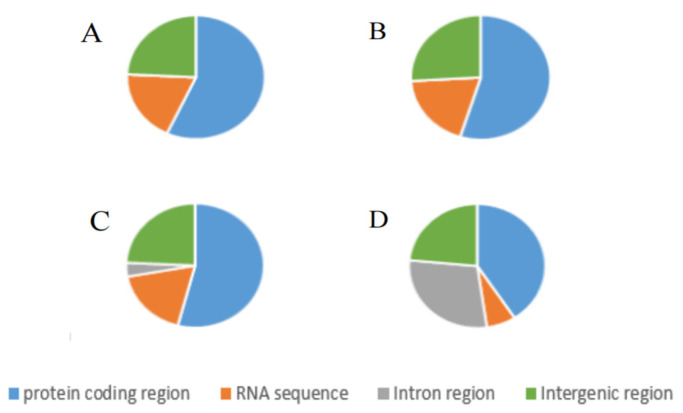
Composition of the three *N. cubana* and one *P. fici* mitochondrial genomes. The pie chart shows the contribution of the different gene regions to the expansion of these four mitogenomes. (**A**) is *N. cubana* WZS16, (**B**) is *N. cubana* BT03, (**C**) is *N. cubana* LD08, and (**D**) is *P. fici* W106-1.

**Figure 3 ijms-25-03093-f003:**
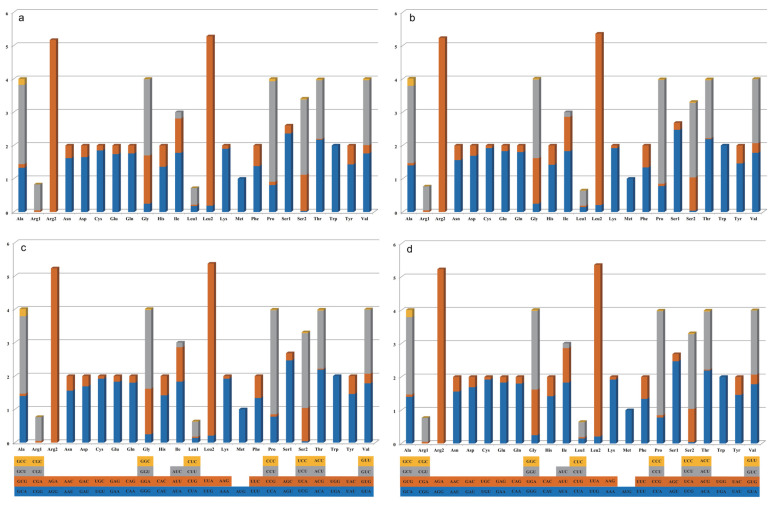
Mitochondrial genome codon usage in three *N. cubana* and *P. fici*. Frequency of codon usage is plotted on the y-axis. (**a**) *N. cubana* WZS16; (**b**) *N. cubana* LD08; (**c**) *N. cubana* BT03; (**d**) *P. fici* W106-1.

**Figure 4 ijms-25-03093-f004:**
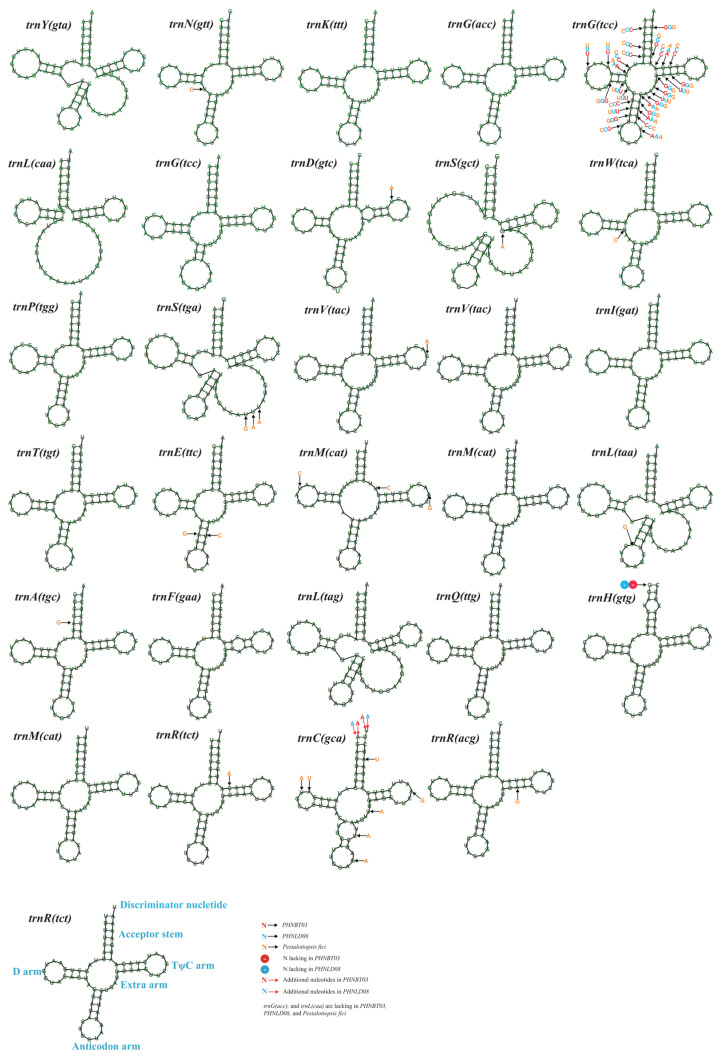
Putative secondary structure of 30 tRNA genes in the mitochondrial genomes of three *N. cubana* and *P. fici* species. Conserved sequences are indicated in green. Variable positions are indicated in red, blue, and orange for BT03, LD08, and W106-1, respectively. Black arrows indicate nucleotide substitutions, the red arrows indicate nucleotide additions, and circles indicate nucleotide deletions. All genes are shown in order of occurrence in the mitochondrial genome, starting with trnY.

**Figure 5 ijms-25-03093-f005:**
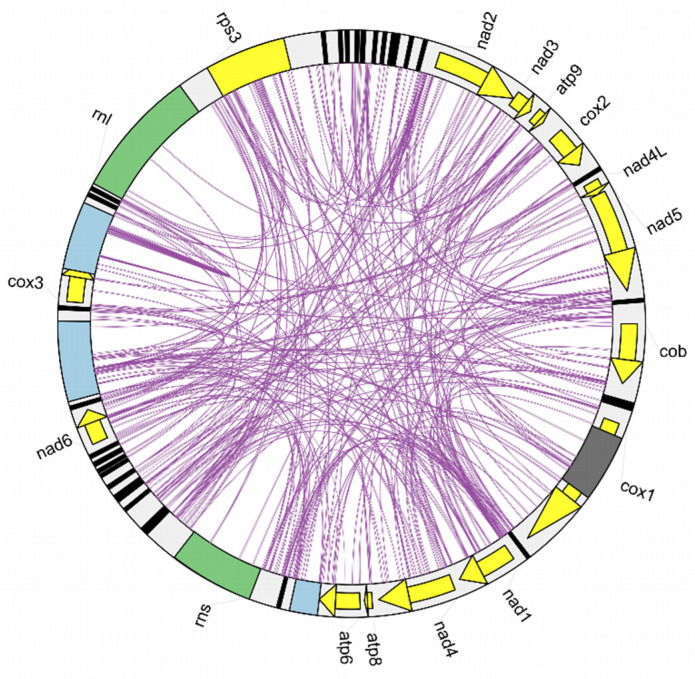
Localization of *N. cubana* LD08 scattered repeat sequences, indicated by the purple line. Arrows indicate core PCGs, grey rectangles are introns, blue rectangles are open reading frames, green rectangles are rrns and rrnL sequences, and all genes are oriented in the same direction.

**Figure 6 ijms-25-03093-f006:**
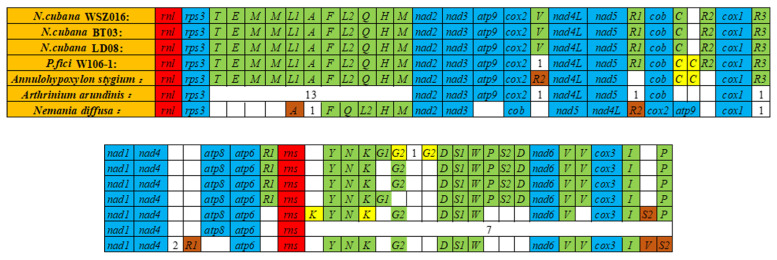
Comparative overview of the gene orientation in the mitogenomes of *N. cubana* isolates and *P. fici*. Core PCGs in all species are in blue; tRNA genes in green; genomic duplication are in yellow; rns and rnl sequences are in red, and tRNA genes possessed by species individually are indicated by their number in a blank box.

**Figure 7 ijms-25-03093-f007:**
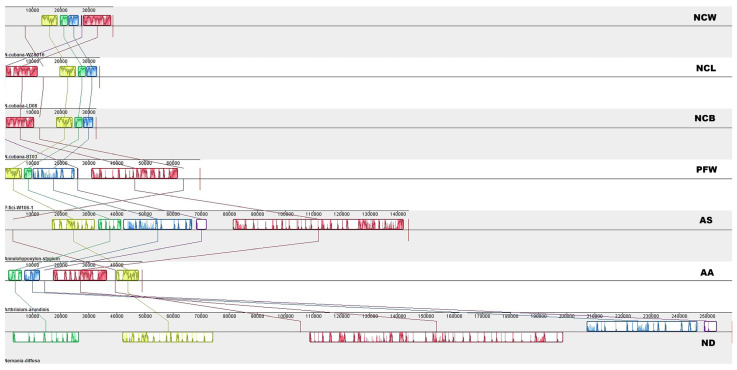
Collinearity analysis of seven mitogenomes from *Xylariales*. Homologous regions between different mitogenomes are represented by blocks of the same color linked by lines. NCW, *N. cubana* WZS16; NCL, *N. cubana* LD08; NCB, *N. cubana* BT03; PFW, *P. fici* W106-1; AS, *Annulohypoxylon stygium*; AA, *Arthrinium arundinis*; ND, *Nemania diffusa*.

**Figure 8 ijms-25-03093-f008:**
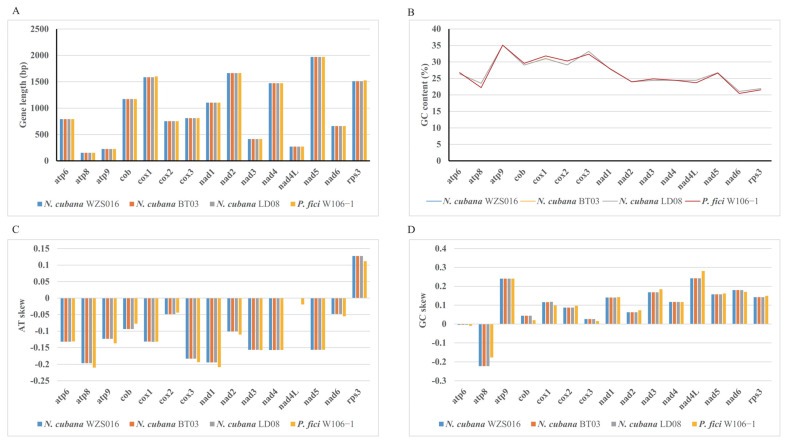
Sequence information of core PCGs in the three *N. cubana* and one *P. fici* species. (**A**) gene lengt. (**B**) GC content. (**C**) AT skew. (**D**) GC skew.

**Figure 9 ijms-25-03093-f009:**
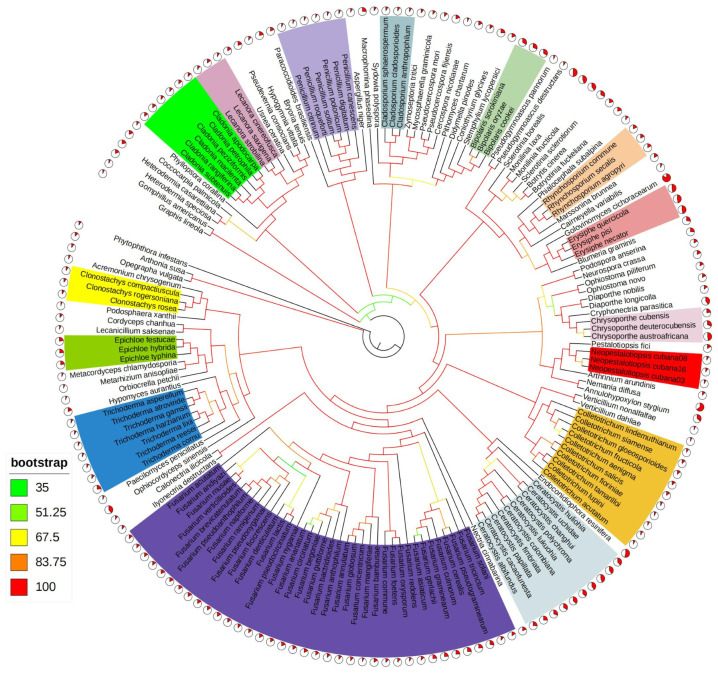
Phylogenetic tree inferred from the tandem mitochondrial genome of core PCGs of 169 *Ascomycetes* based on the maximum likelihood (ML) method. The corresponding colors indicate bootstrap values for tree nodes. Genera containing ≥ 3 species in the species are ascribed the same color. Pie charts represent the total length of the mitochondrial genomes of different species, which is obtained by comparing the length of the mitochondrial genomes of different species with the same length.

**Figure 10 ijms-25-03093-f010:**
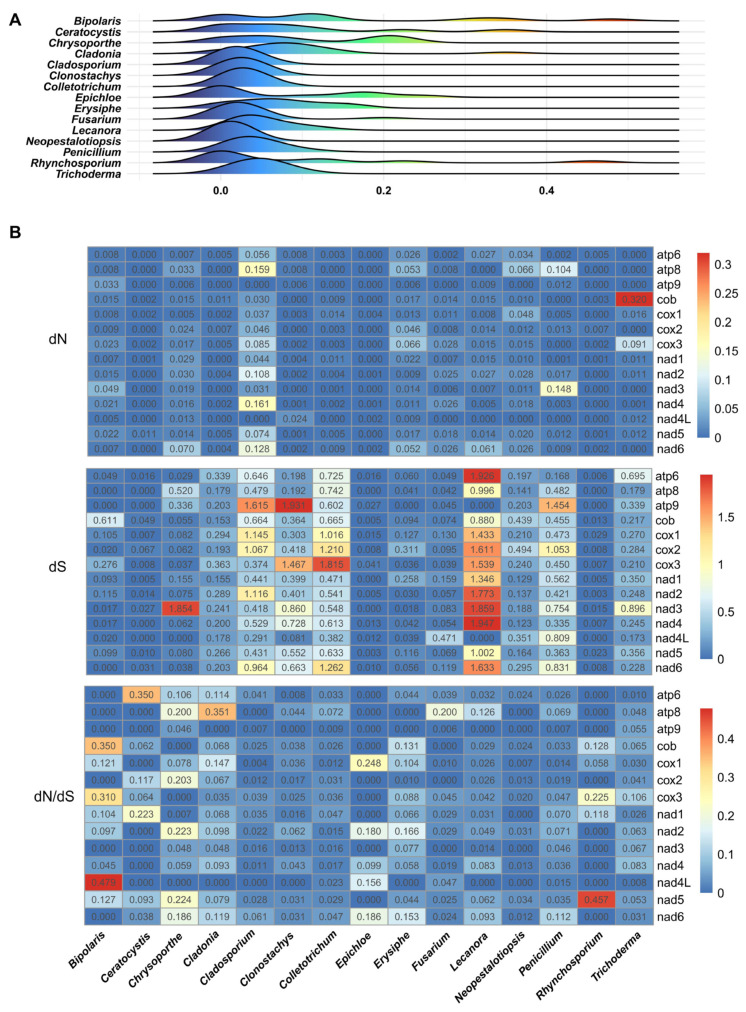
Gene substitution analyses of the fungal core PCGs of the 169 species in the phylum *Ascomycota*; only those with three or more were selected. (**A**) Genus-specific distributions of dN/dS values. The horizontal axis represents dN/dS values. The vertical axis represents logarithmic normal distributions. (**B**) Heatmap shows the dN, dS, and dN/dS values for individual genes.

## Data Availability

Data are contained within the article and Appendix A.

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
