# Peer review of "Comparative Mitogenomics Analysis Revealed Evolutionary Divergence among Neopestalotiopsis Species Complex (Fungi: Xylariales)"

_ijms, 2024, doi:10.3390/ijms25063093_

Round 1

Reviewer 1 Report

Comments and Suggestions for Authors

The paper “Comparative mitogenomics analysis revealed evolutionary divergence among Neopestalotiopsis species complex‘’ aimed to assemble the three complete mitogenomes for three field of guava fruit ring spot pathogen, Neopestalotiopsis cubana.

The paper is prepared professionally. It includes a well-crafted abstract and an exhaustive introduction that justifies the research undertaken. The introduction points to the deficiencies in the literature on the subject. The aim is clearly defined. Modern analytical methods were used in the research. The discussion of the results is well prepared. The conclusions are well-defined. The illustrative material is appropriate.

In my opinion, the manuscript after corrections, will be suitable for publication in a journal.

Detailed comments:

Abstract: Should include some more numeric data obtained from the study

Do not use abbreviations when use first time.

Abstract should be rewritten based on international rules.

Introduction - The introduction is enough in my opinion. Introduction needs some minor changes

Is three N. cubana field isolates is enough for this study???

Reason the recorded variations in intronic sequences in the mitogenome of N. cubana isolates ?????

Figure could be more clearer?????

Comments on the Quality of English Language

NA

Author Response

 School of Tropical Agriculture and Forestry

Key Laboratory of Green Prevention and Control of Tropical Plant Diseases and Pests, Ministry of Education, Hainan University, 58 Renmin Avenue, Meilan District, Haikou, China P.C.:570228

We are pleased to submit a revised version of our original research manuscript entitled “Comparative mitogenomics analysis reveals evolutionary divergence among Neopestalotiopsis species complex” (manuscript number: ijms-2851241) for further appraisal and publication in the esteemed International Journal of Molecular Sciences. We grateful to you and your hardworking team of reviewers for the valuable suggestions and recommendations. We implemented changes recommended by the distinguished reviewers, revised the manuscript, and provided explanations to some concerns raised by the reviewers. The changes effected are tracked in the revised manuscript with blue color and underlined.

All authors have read and consented to the submission of this manuscript. This manuscript has not been published, nor is it under consideration for publication elsewhere. We also declare that there are no competing or conflicting interests. We will be very grateful if our manuscript is considered for peer review to warrant possible publication in International Journal of Molecular Sciences.

Below is a pointed by point response to reviewers' comments;

Sincerely yours,

Wenbo Liu, Prof

School of Tropical Agriculture and Forestry

Hainan University, China

Reviewer #1:

Reviewer comment 1: Abstract: Should include some more numeric data obtained from the study.

Authors Response 1: We have revised the abstract to include numeric values as directed by the reviewer.

Reviewer comment 2: Do not use abbreviations when use first time.

Authors Response 2: We have revised the manuscript and defined all abbreviations at the instance of first usage.

Reviewer comment 3: Abstract should be rewritten based on international rules.

Authors Response 3: We have revised the manuscript to conform with international standards (brief introduction-research problem and objectives-method-results-conclusion).

Reviewer comment 4: Introduction - The introduction is enough in my opinion. Introduction needs some minor changes.

Authors Response 4: We have revised and enhanced the introduction as recommended by the reviewer.

Reviewer comment 5: Is three N. cubana field isolates is enough for this study???

Authors Response 5:  While the three are not enough to support a genera conclusion, it will however, serve as a vital resource to support further association mitogenomic studies that will focus on the collection and sequencing of more field isolates from different geographical locations.

Reviewer comment 6: Reason the recorded variations in intronic sequences in the mitogenome of N. cubana isolates?????

Authors Response 6: We tried to avoid making categorical statements regarding the observed intronic variation because we currently do not have enough data, but our immediate speculation is that the intronic variations recorded between the Ledong isolate and the other two could be an attribute of host selection pressure. We suspected that guava cultivars at Ledong are likely rich and diverse. These multiple resistance attributes will exert enormous selection pressure on associated pathogens, including the Ledong isolates, and could constitute one of the factors accounting for the intronic variations. Also, intensity and long-term exposure to fungicides could trigger genomic and mitogenomic changes to drive species speciation and fungicide resistance. Since the current research has yet to take inventory of all the guava cultivars grown by farmers nor monitored the type and intensity of fungicides used in controlling the fruit scab disease in these areas, we respectfully wish to desist from discussing these results further if the reviewer permits us. These observations have nonetheless opened up a new and pressing research direction.

Reviewer comment 7: Figure could be more clearer?????

Authors Response 7: We enhanced the clarity of figures in the revised manuscript.

Reviewer 2 Report

Comments and Suggestions for Authors

Review on “Comparative mitogenomics analysis revealed evolutionary divergence among Neopestalotiopsis species complex” for manuscript ID ijms-2851241

In this manuscript the authors describe novel mitogenomes of Neopestalotiopsis in comparison with close species. This new knowledge could help to understand evolutionary dynamics of fungal mitogenomes. In the brief introduction authors describe the differences of fungal mitogenomes and closer relatives of Neopestalotiopsis genus.

My questions about Results and Discussion:

Captions on Figure 1 are hard to read, text could be rearranged to make them clearer. What principle used to mutual orientation of circular genomes?

Figure 10B is hard to follow, what authors want to show there?

Methods section comments:

Raw reads are missing.

L92: total DNA was extracted and sequenced, not mtDNA. SPAdes assembler cannot extract mt genome out of total (nuclear) DNA, but NOVOPlasty or GetOrganelle could help.

Annotation of mt genome requires multiple tools, for example MFAnnot is recommended along with MITOS, which is outdated (https://doi.org/10.3389/fpls.2023.1222186).

L125: [34] invalid reference, Mauve paper is https://genome.cshlp.org/content/14/7/1394.abstract

L134: ref [35] isn’t related to MAFFT or alignment methods

ref [36] isn’t related to Interactive Tree of Life.

Author of NC_071220.1 is Wang, W. H. and he is not among the authors of this manuscript. Who is the author of the NCBI submission?

Some minor corrections to the text (style and spelling):

·        L145 invalid accession numbers, “OQ_707026.1” → “OQ707026.1”, “OQ_707025.1” → “OQ707025.1”

·        L14, L136, L138: missing spaces

·        L103: “genetic passwords” → “genetic code” 

·        L118: change reference style (Bernt et al., 2013)

Author Response

 School of Tropical Agriculture and Forestry

Key Laboratory of Green Prevention and Control of Tropical Plant Diseases and Pests, Ministry of Education, Hainan University, 58 Renmin Avenue, Meilan District, Haikou, China P.C.:570228

We are pleased to submit a revised version of our original research manuscript entitled “Comparative mitogenomics analysis reveals evolutionary divergence among Neopestalotiopsis species complex” (manuscript number: ijms-2851241) for further appraisal and publication in the esteemed International Journal of Molecular Sciences. We grateful to you and your hardworking team of reviewers for the valuable suggestions and recommendations. We implemented changes recommended by the distinguished reviewers, revised the manuscript, and provided explanations to some concerns raised by the reviewers. The changes effected are tracked in the revised manuscript with blue color and underlined.

All authors have read and consented to the submission of this manuscript. This manuscript has not been published, nor is it under consideration for publication elsewhere. We also declare that there are no competing or conflicting interests. We will be very grateful if our manuscript is considered for peer review to warrant possible publication in International Journal of Molecular Sciences.

Below is a pointed by point response to reviewers' comments;

Sincerely yours,

Wenbo Liu, Prof

School of Tropical Agriculture and Forestry

Hainan University, China

Reviewer #2:

Reviewer comment 1: Captions on Figure 1 are hard to read, text could be rearranged to make them clearer. What principle used to mutual orientation of circular genomes?

Authors response 1:We have changed the caption of Figure 1 in the manuscript as suggested by the reviewer to make it easier to read. Regarding the principle of mutual localization of circular genomes, the mitogenomes were initially assembled using Spades v3.14.1 software with default parameters with multiple different kmers. Other than screening of mitochondrial contig from resulted assembled sequence, we used the assembled graph file (the gfa file) to retrieve the mitochondrion derived and enclosed topological structure, mainly and manually by Bandage software for visualization and blast-based selection, then traversing the related paths to extract the circular mitochondrial genome.

Reviewer comment 2: Figure 10B is hard to follow, what authors want to show there?

Authors response 2:We have revised the caption of Figure 10 as requested by the reviewer. Regarding the content of Fig. 10B, we added the species P. fici, considering that the sample size of Neopestalotiopsis is small, and the difference in the base content of the core PCGs is almost 0, which is not enough for the calculation of dN/dS values. Since the dN/dS values are relatively small level in the Ascomycota phylum, so we infer that they were mainly subjected to purifying selection during their evolution. We also examined the dN/dS values of other species in the Ascomycota phylum, with the main purpose of exploring the evolutionary differences in the core PCGs of these species during their evolution.

Reviewer comment 3: Raw reads are missing.

Authors response 3:We have added the associated BioProject, SRA and Bio-Sample numbers to Materials and Methods as requested by the reviewers.

Reviewer comment 4: L92: total DNA was extracted and sequenced, not mtDNA. SPAdes assembler cannot extract mt genome out of total (nuclear) DNA, but NOVOPlasty or GetOrganelle could help.

Authors response 4:We have revised the "Materials and Methods" according to the reviewer's request and described the genome assembly procedure in more detail. The mitogenome was initially assembled by Spades v3.14.1 software with default parameters, using multiple different kmers. Other than screening of mitochondrial contig from resulted assembled sequence, we used the assembled graph file (the gfa file) to retrieve the mitochondrion derieved and enclosed topological structure, mainly and manually by Bandage software for visualization and blast-based selection, then traversing the related paths to extract the circular mitochondrial genome.

Reviewer comment 5: Annotation of mt genome requires multiple tools, for example MFAnnot is recommended along with MITOS, which is outdated (https://doi.org/10.3389/fpls.2023.1222186).

Authors response 5:We consulted the relevant references as highlighted by the reviewer and revised the materials and methods in the manuscript and made changes to the [26] references

Reviewer comment 6: L125: [34] invalid reference, Mauve paper is (https://genome.cshlp.org/content/14/7/1394.abstract)

Authors response 6:We revised the [34] references in the revised manuscript Accordingly.

Reviewer comment 7: L134: ref [35] isn’t related to MAFFT or alignment methods

Authors response 7:We revised the [35] references in the revised manuscript.

Reviewer comment 8: ref [36] isn’t related to Interactive Tree of Life.

Authors response 8:We revised the [36] references in the revised manuscript.

Reviewer comment 9: Author of NC_071220.1 is Wang, W. H. and he is not among the authors of this manuscript. Who is the author of the NCBI submission?

Authors response 9:The author of NC_071220.1 is indeed Huanwei Wang, the author of this manuscript, but that he mistyped his name when he uploaded this data.

Reviewer comment 10: L145 invalid accession numbers, “OQ_707026.1” → “OQ707026.1”, “OQ_707025.1” → “OQ707025.1”

Authors response 10:We changed the species accession numbers in the revised manuscript.

Reviewer comment 11: L14, L136, L138: missing spaces

Authors response 11:We have added spaces at the appropriate places in the manuscript as instructed by the reviewers

Reviewer comment 12: L103: “genetic passwords” → “genetic code”

Authors response 12:We changed "genetic passwords" to "genetic code" according to the reviewer's instructions.

Reviewer comment 13: L118: change reference style (Bernt et al., 2013)

Authors response 13:We changed the style of the references as instructed by the reviewers.

Reviewer 3 Report

Comments and Suggestions for Authors

The research article by Huang et al. compared Mitochondrial genome in three Neopestalotiopsis cubana strains along with closely related fungi. The article is surely informative with lots of analysis, but more elaborated organization is needed in writing. There are a few things I would like to point out.   1. Abstract is not clear and concise. The authors should refine the abstract. 2. Line 11- Scab to scab 3. Line 13- filed- why italic? 4. Line 18- Maximum instead of maximum 5. Line 88- Coordinate for the sample collection 6. Line 89- The fungus was isolated and identified- based on what? 7. Throughout the MS- N. cubana instead of the full name. 8. Citation issues such as Line 118 (Bernt et al. 2013) & Line 138 (Minh et al. 2020) 9. Line 131 and throughout the MS- Check letter size or font for P. infestans 10. Line 153- Full name for RPS - firstly introduced 11. Line 188-delete the full name of primary protein coding 12. Line 235- Tandem to tandem 13. Line 258- "("-italic 14. Line 256- Italic for xylariaces 15. Figure 6- Italic for fungi 16. Line 303- Italic for Sporocadaceae 17. Figure 9- need more explanation for the pie chart. Unsure what is the reference genome and how many base pairs? 18. Lin3 391 and 433- references needed 19. The journal put "materials and methods" all the way to the end, so the section should be moved. 20. Line 485-491- It sounds like "conclusion". You can make a seperate "conclusion section" 

Comments on the Quality of English Language

Should go over a few times to refine the manuscript.

Author Response

 School of Tropical Agriculture and Forestry

Key Laboratory of Green Prevention and Control of Tropical Plant Diseases and Pests, Ministry of Education, Hainan University, 58 Renmin Avenue, Meilan District, Haikou, China P.C.:570228

We are pleased to submit a revised version of our original research manuscript entitled “Comparative mitogenomics analysis reveals evolutionary divergence among Neopestalotiopsis species complex” (manuscript number: ijms-2851241) for further appraisal and publication in the esteemed International Journal of Molecular Sciences. We grateful to you and your hardworking team of reviewers for the valuable suggestions and recommendations. We implemented changes recommended by the distinguished reviewers, revised the manuscript, and provided explanations to some concerns raised by the reviewers. The changes effected are tracked in the revised manuscript with blue color and underlined.

All authors have read and consented to the submission of this manuscript. This manuscript has not been published, nor is it under consideration for publication elsewhere. We also declare that there are no competing or conflicting interests. We will be very grateful if our manuscript is considered for peer review to warrant possible publication in International Journal of Molecular Sciences.

Below is a pointed by point response to reviewers' comments;

Sincerely yours,

Wenbo Liu, Prof

School of Tropical Agriculture and Forestry

Hainan University, China

Reviewer #3:

Reviewer comment 1: Abstract is not clear and concise. The authors should refine the abstract.

Authors response 1:We have revised the abstract to make it concise as requested by the reviewers.

Reviewer comment 2: Line 11- Scab to scab

Authors response 2:We corrected the corresponding words in the revised manuscript.

Reviewer comment 3: Line 13- filed- why italic?

Authors response 3:We corrected the corresponding words in the revised manuscript.

Reviewer comment 4: Line 18- Maximum instead of maximum

Authors response 4:We corrected the corresponding words in the revised manuscript.

Reviewer comment 5: Line 88- Coordinate for the sample collection

Authors response 5:We followed the reviewer's instructions to add the geographic location of the collection of the fungi that were mentioned in the revised manuscript.

Reviewer comment 6: Line 89- The fungus was isolated and identified- based on what?

Authors response 6:We have revised the "Materials and Methods" as requested by the reviewer. We collected the fungus from the field, isolated and cultured it, and then determined its pathogenicity by isolating single spores. Then, we observed the conidial morphology of the fungi, and finally, through PCR amplification of three nuclear genes (ITS, TUB2, TEF) and blast comparison in the NCBI database, we confirmed that the three genes of these fungi were all compared to the same species, and thus the results of the co-identification were obtained.

Reviewer comment 7: Throughout the MS- N.cubana instead of the full name.

Authors response 7:We corrected the corresponding words in the revised manuscript.

Reviewer comment 8: Citation issues such as Line 118 (Bernt et al. 2013) & Line 138 (Minh et al. 2020)

Authors response 8:We revised the citation of references in the revised manuscript.

Reviewer comment 9: Line 131 and throughout the MS- Check letter size or font for P. infestans

Authors response 9:We corrected the corresponding words in the revised manuscript.

Reviewer comment 10: Line 153- Full name for RPS - firstly introduced

Authors response 10:We followed the reviewer's instructions for the initial introduction of the words mentioned in the manuscript

Reviewer comment 11: Line 188-delete the full name of primary protein coding

Authors response 11:We deleted words mentioned in the manuscript as instructed by the reviewers

Reviewer comment 12: Line 235- Tandem to tandem

Authors response 12:We corrected the corresponding words in the manuscript as directed by the reviewers

Reviewer comment 13: Line 258- "("-italic

Authors response 13:We corrected the corresponding punctuation in the manuscript as instructed by the reviewers

Reviewer comment 14: Line 256- Italic for xylariaces

Authors response 14:We corrected the corresponding words in the manuscript as directed by the reviewers

Reviewer comment 15: Figure 6- Italic for fungi

Authors response 15:We modified the fungal names in Figure 6 of the manuscript as instructed by the reviewers

Reviewer comment 16: Line 303- Italic for Sporocadaceae

Authors response 16:We corrected the corresponding words in the manuscript as directed by the reviewers

Reviewer comment 17: Figure 9- need more explanation for the pie chart. Unsure what is the reference genome and how many base pairs?

Authors response 17:We revised the caption of Figure 9 in the manuscript as instructed by the reviewers. Regarding the length used for uniform comparison with all species, we set the base length to 400,000 bp

Reviewer comment 18: Lin3 391 and 433- references needed

Authors response 18:We have added relevant [38] references to the Discussion as requested by the reviewers.

Reviewer comment 19: The journal put "materials and methods" all the way to the end, so the section should be moved.

Authors response 19:We have moved the "Materials and Methods" in the manuscript according to the instructions of the reviewers.

Reviewer comment 20: Line 485-491- It sounds like "conclusion". You can make a seperate "conclusion section"

Authors response 20:We appreciate the valuable suggestion but since journal did not create a fromal section for “conclusion” we would not like to add an additional section.

Round 2

Reviewer 2 Report

Comments and Suggestions for Authors

Review on “Comparative mitogenomics analysis revealed evolutionary divergence among Neopestalotiopsis species complex” for manuscript ID ijms-2851241

I would to thank authors for the efforts to improve the manuscript, but it still requires some corrections.

My questions about Results and Discussion:

Quality of Figure 1 became higher, but the layout remained unchanged. I suggest to change the sans serif font.

Figure 10B might be redrawn from scratch. Now this figure contains 15 taxa, but most of them weren’t considered in text except L345-347. Figure 10B should illustrate the authors’ point L336-343.

Methods section comments:

L509: mean reads length of NovaSeq (150) doesn’t fit the libraries (350)

Assembly details doesn’t explain how the mt-genome contigs were obtained from the total DNA assembly (L514), what reference genome was used to arrange the contigs? How the whole circular chromosomes were able to be built when the raw reads contain a lot of bacterial reads? SRR23999140 has bacterial contamination https://trace.ncbi.nlm.nih.gov/Traces/?view=run_browser&acc=SRR23999140&display=analysis

Accession SRR20075021 mostly contains bacterial reads, see NCBI analysis via URL https://trace.ncbi.nlm.nih.gov/Traces/?view=run_browser&acc=SRR20075021&display=analysis

Bandage software requires the reference.

Simply adding MFannot reference to the manuscript is not enough, while the genomes weren’t re-annotated.

Some minor corrections to the text (style and spelling):

·        L495“blast” → “BLAST”

·        L515: “mitos” → “MITOS”

·        L519: “blastn”→”BLASTn”

·        L568: remove duplicates

Author Response

 School of Tropical Agriculture and Forestry

We are pleased to submit a revised version of our original research manuscript entitled “Comparative mitogenomics analysis reveals evolutionary divergence among Neopestalotiopsis species complex” (manuscript number: ijms-2851241) for further appraisal and publication in the esteemed International Journal of Molecular Sciences. We grateful to you and your hardworking team of reviewers for the valuable suggestions and recommendations. We implemented changes recommended by the distinguished reviewers, revised the manuscript, and provided explanations to some concerns raised by the reviewers. The changes effected are tracked in the revised manuscript with red color and underlined.

All authors have read and consented to the submission of this manuscript. This manuscript has not been published, nor is it under consideration for publication elsewhere. We also declare that there are no competing or conflicting interests. We will be very grateful if our manuscript is considered for peer review to warrant possible publication in International Journal of Molecular Sciences.

Below is a pointed by point response to reviewers' comments;

Sincerely yours,

Wenbo Liu, Prof

School of Tropical Agriculture and Forestry

Hainan University, China

Reviewer #2:

Reviewer comment 1: Quality of Figure 1 became higher, but the layout remained unchanged. I suggest to change the sans serif font.

Authors response 1:We changed the font of Figure 1 to a sans-serif font as requested by the reviewer

Reviewer comment 2: Figure 10B might be redrawn from scratch. Now this figure contains 15 taxa, but most of them weren’t considered in text except L345-347. Figure 10B should illustrate the authors’ point L336-343.

Authors response 2:We followed the reviewer's request and firstly made some adjustments to Figure 10B and re-described the values of dN and dS for most of the taxa, and finally added some author's points.

Reviewer comment 3: L509: mean reads length of NovaSeq (150) doesn’t fit the libraries (350)

Authors response 3:We previously constructed a 350bp library, the actual insert fragment is 300bp, the other 50bp is double-ended primer and index, and the reads output reads of 150bp in length.

Reviewer comment 4: Assembly details doesn’t explain how the mt-genome contigs were obtained from the total DNA assembly (L514), what reference genome was used to arrange the contigs? How the whole circular chromosomes were able to be built when the raw reads contain a lot of bacterial reads? SRR23999140 has bacterial contamination https://trace.ncbi.nlm.nih.gov/Traces/?view=run_browser&acc=SRR23999140&display=analysis

Accession SRR20075021 mostly contains bacterial reads, see NCBI analysis via URL https://trace.ncbi.nlm.nih.gov/Traces/?view=run_browser&acc=SRR20075021&display=analysis

Authors response 4:This mitogenome was initially assembled by Spades v3.14.1 software with default parameters, using multiple different kmers. The contig assembled by Spades de novo included bacterial sequences, but we screened for mitochondrial contigs from the assembled sequences using Neopestalotiopsis cubana (NC_071220), we used the assembled graph file (the gfa file) to retrieve the mitochondrion derieved and enclosed topological structure, mainly and manually by Bandage software for visualization and blast-based selection, then traversing the related paths to extract the circular mitochondrial genome.

Reviewer comment 5: Bandage software requires the reference.

Authors response 5:We have added [26] references as requested by the reviewers

Reviewer comment 6: Simply adding MFannot reference to the manuscript is not enough, while the genomes weren’t re-annotated.

Authors response 6:We have previously annotated these mitochondrial genomes with MFannot, except that the software was omitted from the formulation.

Reviewer comment 7: L495“blast” → “BLAST”

Authors response 7:We changed the word "blast" to "BLAST" as requested by the reviewer.

Reviewer comment 8: L515: “mitos” → “MITOS”

Authors response 8:We changed "mitos" to "MITOS" as requested by the reviewer.

Reviewer comment 10: L519: “blastn” → “BLASTn”

Authors response 10:We changed "blastn" to "BLASTn" as requested by the reviewer.

Reviewer comment 11: L568: remove duplicates

Authors response 11:We have removed the duplicate "PRJNA857506" as requested by the reviewer.

Reviewer 3 Report

Comments and Suggestions for Authors

The authors addressed all of my concerns and made great progress toward acceptance. 

Author Response

we are grateful to the reviewer for approving out manuscript

Round 3

Reviewer 2 Report

Comments and Suggestions for Authors

Review on “Comparative mitogenomics analysis revealed evolutionary divergence among Neopestalotiopsis species complex” for manuscript ID ijms-2851241

I would to thank authors for the improving figures and the manuscript, but some concerns remain unclear.

My questions about Results and Discussion:

As I can understand, Figure 10B should show the purifying selection intensity among several genera. The approach to illustrate this point by various colors and line types lead to unclear drawing. The heatmap, where color scale shows the value of dN, dS and dN/dS (instead the length of the line) might look better. The rows could stand for the genes, and columns for the taxa.

Methods section comments:

My comment remained unanswered. How the whole circular chromosomes were able to be built when the raw reads contain mostly of bacterial reads? Among all raw identified reads SRR23999140 contains less than 1% of eukaryotic ones https://trace.ncbi.nlm.nih.gov/Traces/?view=run_browser&acc=SRR23999140&display=analysis

Some minor corrections to the text (style and spelling): there are some missing spaces (L15, L438) and different font size.

Author Response

 School of Tropical Agriculture and Forestry

We are pleased to submit a revised version of our original research manuscript entitled “Comparative mitogenomics analysis reveals evolutionary divergence among Neopestalotiopsis species complex” (manuscript number: ijms-2851241) for further appraisal and publication in the esteemed International Journal of Molecular Sciences. We grateful to you and your hardworking team of reviewers for the valuable suggestions and recommendations. We implemented changes recommended by the distinguished reviewers, revised the manuscript, and provided explanations to some concerns raised by the reviewers. The changes effected are tracked in the revised manuscript with red color and underlined.

All authors have read and consented to the submission of this manuscript. This manuscript has not been published, nor is it under consideration for publication elsewhere. We also declare that there are no competing or conflicting interests. We will be very grateful if our manuscript is considered for peer review to warrant possible publication in International Journal of Molecular Sciences.

Below is a pointed by point response to reviewers' comments;

Sincerely yours,

Wenbo Liu, Prof

School of Tropical Agriculture and Forestry

Hainan University, China

Reviewer #2:

Reviewer comment 1: As I can understand, Figure 10B should show the purifying selection intensity among several genera. The approach to illustrate this point by various colors and line types lead to unclear drawing. The heatmap, where color scale shows the value of dN, dS and dN/dS (instead the length of the line) might look better. The rows could stand for the genes, and columns for the taxa.

Authors response 1:We are glad that the reviewers were able to provide comments for the figure of the manuscript, and have revised Figure 10B in the manuscript. We modified the dot plot to a heat map and showed the specific values of dN versus dS, with columns representing genes and rows representing taxa.

Reviewer comment 2: My comment remained unanswered. How the whole circular chromosomes were able to be built when the raw reads contain mostly of bacterial reads? Among all raw identified reads SRR23999140 contains less than 1% of eukaryotic ones https://trace.ncbi.nlm.nih.gov/Traces/?view=run_browser&acc=SRR23999140&display=analysis

Authors response 2:Thank you for raising this question, which is indeed a crucial issue in the assembly of organelle genomes. Essentially, the sequencing data obtained from total DNA not only contains organelle data but also includes nuclear DNA data and even some level of exogenous DNA contamination, such as bacteria. Currently, in the process of organelle genome assembly, these non-targeted reads can be excluded. There are two main strategies: one is to directly obtain a set of reads through a bait approach using a seed sequence (usually a fragment of organelles from related species), and then assemble the complete organelle genome. However, this method carries a risk of incomplete assembly, often due to inadequate baiting or the seed sequence being too distantly related. The second method involves assembling the total DNA data and obtaining the target graph from the assembled graph using similarity alignment. In this study, the mitochondrial graph was obtained using this method. Typically, the mitochondrial graph is a closed circular graph or a closed unresolved cluster of paths. It can also be visually identified using sequencing depth filtering, such as with the Bandage tool. This is because the copy number of mitochondria is typically several orders of magnitude higher than that of contamination or nuclear DNA, making them easily distinguishable. The illumina short sequences were aligned with the mitochondrial sequences using the BWA software, followed by the coverage statistics using samtools depth, and finally the coverage graph was drawn using python script. The horizontal coordinate is the length of the mitochondrial genome, and the vertical coordinate is the coverage of the mitochondrial genome in the sequencing data. From the graph, we can see that the coverage of the mitochondrial genome spliced from the sequencing data corresponding to SRR23999140 is as high as 7638.39X, and the coverage of the lowest position is as high as 4303X, indicating that there are a lot of mitochondrial reads in the sequencing data, and the quality of the assembled data is very reliable. data quality is very reliable.

Reviewer comment 3: Some minor corrections to the text (style and spelling): there are some missing spaces (L15, L438) and different font size.

Authors response 3:We have adjusted the font as requested by the reviewer.

sequencing depth
